# Microbiota and Immunity during Respiratory Infections: Lung and Gut Affair

**DOI:** 10.3390/ijms25074051

**Published:** 2024-04-05

**Authors:** Veronica Marrella, Federico Nicchiotti, Barbara Cassani

**Affiliations:** 1UOS Milan Unit, Istituto di Ricerca Genetica e Biomedica (IRGB), CNR, 20138 Milan, Italy; veronica.marrella@humanitasresearch.it; 2IRCCS Humanitas Research Hospital, 20089 Milan, Italy; 3Department of Medical Biotechnologies and Translational Medicine, Università degli Studi di Milano, 20089 Milan, Italy; federico.nicchiotti@gmail.com

**Keywords:** microbiota, dysbiosis, mucosal immunity, host–microbe interactions, gut–lung axis, microbial therapies, bacteria, virus

## Abstract

Bacterial and viral respiratory tract infections are the most common infectious diseases, leading to worldwide morbidity and mortality. In the past 10 years, the importance of lung microbiota emerged in the context of pulmonary diseases, although the mechanisms by which it impacts the intestinal environment have not yet been fully identified. On the contrary, gut microbial dysbiosis is associated with disease etiology or/and development in the lung. In this review, we present an overview of the lung microbiome modifications occurring during respiratory infections, namely, reduced community diversity and increased microbial burden, and of the downstream consequences on host–pathogen interaction, inflammatory signals, and cytokines production, in turn affecting the disease progression and outcome. Particularly, we focus on the role of the gut–lung bidirectional communication in shaping inflammation and immunity in this context, resuming both animal and human studies. Moreover, we discuss the challenges and possibilities related to novel microbial-based (probiotics and dietary supplementation) and microbial-targeted therapies (antibacterial monoclonal antibodies and bacteriophages), aimed to remodel the composition of resident microbial communities and restore health. Finally, we propose an outlook of some relevant questions in the field to be answered with future research, which may have translational relevance for the prevention and control of respiratory infections.

## 1. Introduction

Respiratory tract infections are the most common category of infectious diseases and one of the leading causes of morbidity and mortality worldwide, inflicting social and economic burden [1,2]. Thanks to the technological advancement of the last decade, the healthy lung, previously regarded as a sterile organ, is now described to harbor its own specific microbial population [3]. Lung microbiota is considered a transient settlement of bacteria continuously inhaled and eliminated [4]. These commensal bacteria act on the immune system, inducing protective response and preventing invasion and colonization by pathogens. At the same time, they directly inhibit the growth of pathogens through the production of anti-microbial products. Indeed, a continuous dialog between commensal bacteria and resident epithelial and immune cells support the lung homeostasis [5]. Accordingly, the lung microbiota is regarded as a “mirror of lung health status”: several studies indicate that the lung bacterial composition as well as the lung environment drastically change during the occurrence of pulmonary pathologies [6]. On the other hand, it has become evident that the lung is engaged in a continuous bidirectional cross-talk with the gut and that altered microbiota composition at either site can contribute to the development and progression of distal diseases [7].

Here, we will focus on the role of host–microbiota interaction in health and during most common respiratory bacterial (*Mycobacterium tubercolosis*, *Streptococcus pneumoniae*, *Klebsiella pneumoniae*, and *Haemophilus influenza*) and viral (Influenza virus, Respiratory syncytial virus and Severe Acute Respiratory Syndrome Coronavirus 2) infections, describing the lung physiological defenses, their involvement, and their changes in disease setting. Specifically, we will highlight the importance of the gut–lung axis, involving the exchange of microbes, metabolites, and inflammatory mediators as well as immune cell trafficking, in lung disease progression and outcome.

Finally, we will review the most important knowledge regarding the use of microbiota-based therapeutic approaches for the treatment of respiratory infections.

## 2. Lung Microbiota

Technical advances in next generation sequencing and independent validation from worldwide laboratories set the stage to delineate the form and function of the lung microbiome [3]. A large number of studies have demonstrated that the healthy lungs harbor a microbiota enriched in the phyla *Bacteroidetes* and *Firmicutes*, with *Prevotella*, *Streptococcus*, *Veilonella*, *Neisseria*, *Haemophilus*, and *Fusobacterium* being the most abundant genera [8,9,10,11] (Figure 1). The pulmonary environment is generally inhospitable for bacterial community development, resulting in relatively low bacterial replication rate and low microbial biomass compared to the intestinal one (10^3^–10^5^ versus 10^11^–10^12^ bacteria/gram tissue). In healthy individuals, the upper respiratory tract (URT) includes a large and complex microbiome in which oral commensal taxa are prevalent. In contrast, the microbiome of the lower respiratory tract (LRT) has generally a rather low biomass, defined by concurrent dynamics of importation, via microaspiration from the URT, and elimination, via mucociliary activity and innate immune function, with limited local microbial replication. Microbial immigration occurs through inhalation of airborne bacteria, direct dispersion along mucosal surfaces, and microaspiration, with the latter considered the dominant and ubiquitous route among healthy subjects. Indeed, the bacterial community of the LRT largely resembles the oral microbiota composition [12,13]. The dynamic nature of the lung microbiome might be an important distinctive property as compared to the microbial behavior of high biomass mucosae (i.e., oral cavity, gut) where microbial communities are highly resistant [7,14,15]. The 16*S* rRNA profiling has identified spatial variation in respiratory microbiota between healthy URT and LRT. A bacterial overlap with *Streptococcus* genera was found continually from the oral cavity to the lower lung. However, *Staphylococci* usually inhabit the upper airways, whereas *Prevotella* and *Veillonella* prevail in the lower airways [12,16].

Colonization of the airways starts immediately at birth within 24 h of life [17]. In the healthy condition, species from the *Streptococcus*, *Staphylococcus*, *Prevotella*, *Moraxella*, *Haemophilus*, *Lactobacillus*, *Corynebacterium*, and *Dolosigranulum* genera represent the initial colonizers of the respiratory tract early in life and contribute to a balanced and dynamic microbial community [18]. The bacterial load in the respiratory tract increases from neonatal age to maturity, by expanding diversity and functional capacity with many influencing factors, including maternal delivery, breastfeeding, antibiotic usage, and smoke exposure. Studies in animal models demonstrated bacterial load increasing over the first 2 weeks of life with phyla shifting from *Gammaproteobacteria* and *Firmicutes* towards *Bacteroidetes* [19]. The term “dysbiosis” indicates changes in the balance of the three determinants of a healthy lung microbiome (microbial immigration, elimination of microbes, and resident reproduction rate) and more generally accounts for a disturbance of the host microbial ecosystem. Dysbiosis in the lung is associated with many adverse biological events and participates in the development and progression of respiratory diseases [20,21]. Ph, temperature, oxygen tension, and nutrient availability are among the ecological factors shaping the growth conditions of the lung microbiome, together with local microbial competition, host epithelial cell interactions, and inflammatory cell activation. Indeed, regional growing conditions of the lung microbiome are dramatically altered during pathology, resulting in disease- and patient-proper microbial communities suitable for the injured airways [3,20,22]. In the disease condition, the presence and abundance of pathogens in the lung are not consistently mirrored in the oropharynx, underlining the importance of direct sampling of the lower airway with bronchoalveolar lavage (BALF) in investigating lung microbiota [23]. Several studies comparing diseased versus healthy lungs found significant differences in community composition, with the disease condition being associated with loss of bacteria diversity or with a dominance of a single taxon or small group of taxa [24,25,26]. Microbial dysbiosis characterizes various lung diseases, in which a reduced microbial diversity has been also associated with disease progression [7,27,28,29].

On the other hand, it is not still clear if microbial dysbiosis itself is the origin or the result of the disease. For instance, pathophysiological changes in lung architecture and impaired mucus clearance might result in microbial dysbiosis instead of being caused by it. Another possibility is that dysbiosis may give rise to the disease inducing inflammatory signals (such as NF-κB, Ras, IL-17, and PI3K) or shutting down of tumor necrosis factor (TNF) and interferon (IFN) γ production, in response to lower airways’ pathogens [30,31,32]. Enrichment of microbial oral community within the lung, including anaerobes such as *Prevotella* and *Veillonella*, was associated with high inflammatory state and increased infection susceptibility due to altered immune response. Additionally, the altered composition of the lung microbiota poses a greater risk for some individuals of acquiring infections [29,33]. Lung microbiota modifications have been implicated in the worsening of several pulmonary diseases, and different cellular immune responses are associated with exposure to various lung microbes. Indeed, in animal models of chronic lung inflammation or of *pneumotype SPT*, enrichment of *Pseudomonas* and *Lactobacillus*, derived from a pathological human bronchoalveolar system, correlates to an enhanced Th17 type response [30,34]. Pathobionts such as members of *Proteobacteria* induce severe Toll-like receptor (TLR)2-independent airway inflammation and lung immunopathology [35]. Respiratory tract microbial composition influences baseline inflammation in humans and mice: microbiota can affect IL-17 production by pulmonary γδ T cells [36] and differentiation of alveolar macrophages (AM) [37]. Microbiota also influence the ability to fight against respiratory infection. *Haemophilus parainfluenzae* enables TLR4 to activate pro-inflammatory response, to hinder the corticosteroid-related pathway and to induce inflammatory Th2 pathways, finally leading to bronchial high responsiveness [38]. Notably, the make-up of the lung microbiota is influenced, as shown for the gut, by different factors among which are diet, body mass index (BMI), and gender [39,40]. Nevertheless, the study of the lung microbiome, similar to the interplay between commensal microbial communities and pulmonary immunity, is still at the beginning, with future studies needed to deeply uncover the specific mechanisms involved.

### 2.1. Lung Microbiota Modifications during Respiratory Infections

The origin of lung infections includes the acquisition of a pathogen, its spread, and invasion into the lower respiratory tract [41].

Pneumonia is an inflammation of the lung parenchyma. Etiologically, it is classified as community-acquired pneumonia (CAP infection in a previously healthy individual) or hospital-acquired pneumonia (HAP infection in a hospitalized individual within 48 h of admission). The Gram-positive *Streptococcus* (*S.*) *pneumoniae* is responsible for the majority of CAP cases and a leading cause of illness in children under 2 years, in aged people, and in immunocompromised individuals. Pneumonias due to Gram-negative *Haemophylus* (*H.*) *influenzae* and *Klebsiella* (*K.*) *pneumoniae* spread among patients over 50 years with chronic obstructive lung disease or alcoholism, while pneumonias due to Gram-negative *Mycoplasma pneumoniae* and *Chlamydophila pneumoniae* are more diffuse in children. Viral pneumonias caused by respiratory syncytial virus (RSV) and adenoviruses are rare in the healthy population, whereas pneumonia caused by influenza viruses is still a cause of high mortality in old people and in patients with underlying diseases [42]. Ecologically, infections are characterized by an increased microbial burden and a reduction in community diversity, together with an increased host inflammation and tissue injury [43]. Alterations in the microbiome of the URT are responsible for the occurrence of bacterial infections leading to pneumonia. Pharingeal colonization by Gram-negative *S. pneumoniae*, *H. influenzae*, or *Moraxella cataralis* in healthy neonates is associated with higher risk of developing broncholitis [44]. In adults, the predominance of *S. Pneumonia*, *Lactobacilli*, and *Rothia* was associated with pneumonia [15]. Viral infections have a bidirectional relationship with the respiratory microbiome: abundance of *Streptococcus*. and *Prevotella salvia* associated with reduced influenza A development in exposed individuals [45]. On the other hand, *Pseudomonas* was found increased in influenza patients [46]. Human studies showed that early colonization with different strains was associated with increased risk of developing respiratory infections [47,48] (Figure 1). Accordingly, both pathogen co-viral infection and microbial interactions may influence the course of disease.

A recent work describes a significant increased viral replication in severe pulmonary infections, and a difference in microbial interactions between patients with bland and severe disease, especially the association between the common pathogenic bacteria and *Rothia* [26]. The high mortality risk due to influenza is mostly attributed to secondary bacterial infections. Viruses increase host vulnerability to bacterial colonization through several mechanisms, but the role of the host micro-environment in LRT infected with influenza virus (IV) is poorly described. The LRT microbiome, lung transcriptome, and BALF metabolome in mice inoculated intra-nasally with H1N1 virus, to simulate human influenza, showed important modifications, which were maintained in the recovery period. This suggests that IV infection generated a long-term effect in LRT micro-environmental homeostasis, beneficial for potential pathogens [49].

The Gram-variable *Mycobacterium tuberculosis* (*Mtb)* is responsible for pulmonary pneumonia in the tuberculosis (TB) disease. Studies performed both in mice and humans demonstrate that the pulmonary microbiome plays a role in resistance to *Mtb* infections [50,51]. Indeed, it is associated with various states of TB, with abundance of *Pseudomonas* being associated with increased risk of treatment failure [52]. Individuals infected with *Mtb* have reduced microbiota diversity compared to healthy controls and often show enrichment of *Streptococcus* and *Pseudomonas* [53,54,55]. Studies highlight that some bacterial strains may be associated with TB onset, its recurrence, and therapy failure [56,57]. In TB, the lung and the entire LRT present with peculiar microbial features: *Mtb*, *Staphylococcus aureus*, and *Kluyveromyces lactis* were highly represented in a collection of BALF lavage fluid from 123 patients with TB, whereas *H. Parainfluenzae* was enriched in uninfected lungs [58]. A study reported that TB patients showed reduced diversity of alveolar microbiota (reduction in *Streptococcus* and *Fusobacterium* and an increase in *Mtb* abundance) compared with healthy controls. These changes might be determined by the inflammatory environment, as *Mtb* could release virulence factors, which suppressed the macrophage response [59]. In another study, *Streptococcus* was significantly increased in TB, and the Th1-response in TB group may be triggered by *Neisseria* and *Haemophilus* [60]. A survey using sputum samples from India indicated *Neisseria* and *Veillonella* as two dominant genera in the TB group, together with the opportunistic pathogen *Rothia mucilaginosa*, known to have a widespread diffusion in TB patients [57]. A study using nasopharyngeal swab samples revealed that *Mtb* infection significantly changed microbiota composition: *Proteobacteria*, *Gammaproteobacteria*, *Pseudomonadales*, and *Moraxellaceae* increased, while phyla *Bacillales* and *Lachnospiraceae* decreased in TB patients compared to healthy controls. Moreover, the reduced presence of *Corynebacterium* in the TB group may be related to nutritional status, immune-related molecules, and inflammation-related markers [61].

In 2019, a novel coronavirus Severe Acute Respiratory Syndrome 2 (SARS-CoV-2) has caused the coronavirus disease (COVID-19) pandemic. The disease severity and mortality rate differed based on age comorbidities, many of which were linked to gut and lung microbial alterations. This suggests that dysbiosis can predict COVID-19 severity [62]. Generally, the lungs of critically ill COVID-19 patients likely have reduced species diversity and increased bacterial burden, compared to healthy or less sick COVID-19 individuals [63]. Poor clinical outcome was associated with lower airway enrichment of commensal *Mycoplasma salivarium* [64]. A study described BALF samples from severe COVID-19 patients with a significant higher abundance of pseudomonas, while BALF from COVID-19 pneumonia-negative patients were characterized by enrichment of *H. Influenzae* and *Veillonella dispar* among others [65]. The meta-transcriptomic analysis of BALF showed an important expansion of bacterial load and other pathogens, underlining the presence of lung dysbiosis in COVID-19 [66]. In line with this, it has been reported that the microbiome of lung tissue in 20 deceased COVID-19 patients was dominated by *Acinetobacter*, commonly connected to lung infections causing pneumonia [67]. On the other hand, an analysis of the nasopharyngeal microbiomes from patients suffering from an acute respiratory illness revealed no difference in the composition or diversity, when comparing patients confirmed to have COVID-19 with the negative ones [68]. Similarly, Minich et al. reported that common phyla in nasopharyngeal samples, regardless of COVID-19, (i.e., *Firmicutes*, *Actinobacteria*, *Bacteroidetes*, and *Proteobacteria*) were predominant also in COVID-19 patients, suggesting that SARS-CoV-2 infection does not importantly alter the microbiome from a healthy condition [69]. Through a meta-transcriptomic analysis on COVID-19 and control swab specimens, the upper airway dysbiosis was characterized with a *Streptococcus*-dominant microbiota specifically present in COVID-19 patients [70]. In conclusion, the diversity of respiratory microbiota in COVID-19 patients remains a debated issue [71].

### 2.2. Lung Immune Responses and Host–Microbe Interactions during Respiratory Infections

In URT and LRT, the host immune system responds against potentially deleterious agents and distinguishes them from self-components, foreign non-dangerous material, and beneficial commensal microbiota. A complex network of local epithelial and immune cells is responsible for maintaining lung homeostasis [72] (Figure 1). The airways epithelium is a biophysical protective barrier and a site of interaction with the local microbiota [4,15,73]. Secretory cells of the airway epithelium that produce mucus are a key element of the lung’s innate immune system [74]. Several data suggest that respiratory microbiota regulate the epithelial barrier shaping mucus production. Specifically, in mouse models, a link between mucins and response to pulmonary infections was demonstrated [75,76].

In addition to mucins, the airway epithelial cells provide an antigen-specific secretory IgA barrier able to protect the airway surface [77]. Secretory IgA functions by preventing the adsorption of pathogens, neutralizing their toxic products at the mucosal epithelium, mediating virus elimination in infected epithelial cells, and promoting the killing of pathogens [78]. Several studies have demonstrated a role for IgA in the defense strategies against respiratory infections: IgA-deficient mice exhibit increased susceptibility to intranasal infection with *Mycobacterium bovis* Bacillo di Calmette-Guérin (BCG) [79]. Intranasal administration of neutralizing IgA, followed by RSV, influenza virus (IV) or reovirus challenge, resulted in a meaningful decrease in pulmonary viral titer and reduced pneumonia severity in murine models. Notably, intravenous administration of antigen-specific polymeric IgA protected the mice from influenza infection due to the nasal secretion of IgA from serum [80].

Local airways also secrete protective mediators called Antimicrobial Peptides (AMPs), including lysozyme, lactoferrin, lipocalins, peroxidase, aminopeptidases, collectins (surfactant protein A and surfactant protein D), mannan-binding lectin (MBL), cathelicidins, and β-defensins [81]. Notably, β-defensins and cathelicidins show antimicrobial and immunomodulatory effects and are involved in shaping the microbiota composition. Indeed, the direct antimicrobial activity and immunomodulation of inflammatory responses is triggered by members of the microbiota [78].

Different types of innate immune cells are present in the lung: alveolar macrophages (AMs) are the most representative ones and appear to be central in the pathogenesis of several respiratory tract infections, including *Mtb*, *S. pneumoniae*, rhinovirus, IV, and RSV [82]. AMs start the leukocyte recruitment and directly eliminate the pathogen using several pathogen-specific mechanisms, such as secretion of pro-inflammatory cytokines/chemokines (IL-6, IL-8, or CXCL10), initiation of type I IFN signaling, enhanced expression of pattern recognition receptors (PRR), together with inhibition of nuclear export of viral genome [83]. Increased morbidity and mortality were associated with AM depletion both experimentally and during the natural course respiratory viral infection [84,85]. Accordingly, a consistent feature of severe COVID-19 is dysregulation of pulmonary macrophages [86].

Dendritic cells (DCs) in the lungs have a role in protection against respiratory infections, mounting a robust adaptive immune response towards pathogens. DCs can initiate the antiviral CD8 cytotoxic T-cell responses that leads to viral clearance and also control the level of inflammatory responses, contributing significantly to the severity of disease [87]. DC alterations during and after inflammation can be employed as biomarkers of susceptibility to secondary pneumonia, as well as promising therapeutic targets to enhance outcomes in patients [88]. In *Mtb* infections, DCs play multifactorial roles in shifting innate immune responses to adaptive immunity [89]. Notably, DCs have key roles in defense against SARS-CoV-2 infection [90].

Innate lymphoid cells (ILCs) mediate protective immunity from pathogens and parasites and promote tissue repair and homeostasis following infections, but their altered functions may also contribute to pathogenesis [91,92]. A recent paper determines the role of ILC3 in the early immune events necessary to achieve immune protection during *Mtb* infection [93]. An observational study suggests that, by promoting disease tolerance, homeostatic ILCs decrease morbidity and mortality associated with SARS-CoV-2 infection, and that reduced number of ILCs contribute to increased COVID-19 severity with age [94]. Mucosal-associated invariant T (MAIT) cells are a subset of unconventional T cells that carry out immune-surveillance and immunity against microbial infection [95,96]. Different studies proposed a role for MAIT cells in the immune control of *Mtb* infection by recognition and killing of cells infected by bacterium, including DCs and pulmonary epithelial cells [95]. Moreover, cytokines are required for MAIT cells to respond to *Mtb* antigens both in human and mice, which may be due to their recruitment at the infected site [97,98]. A new study confirms that circulating MAITs are activated but reduced in frequency in patients with acute SARS-CoV-2 infection, whereas they accumulate in the lungs of individuals with lethal COVID-19 [99].

Invariant Natural Killer T (iNKT) cells have a role in the control of commensals, including opportunistic pathogenic microbiota, and reciprocally, the microbiota regulates iNKT cells. During primary RSV infection, activation of lung iNKT cells leads to antiviral CD8 T-lymphocyte response and virus clearance, in addition to causing pulmonary eosinophilia and fibrosis. In murine infections, intranasal administration of α-GalCer can prophylactically protect against lethal *S. pneumoniae* infection and defend susceptible mice from *Mtb* [100]. iNKT cell deficiency in humans could be fundamental for the development of active/acute TB: patients with active TB have less peripheral iNKT cells compared with those with latent TB, and normal iNKT cell frequencies can be re-established by treatment for active TB [101]. A reduction in iNKT cells in patients with severe COVID-19 pneumonia was described, suggesting a potential role of this subset as a biomarker of the severity of the disease [102].

Lung resident memory T (TRM) cells, mostly differentiated from effector T cells, create specific niches and stay lastingly in lung tissues. When infection re-occurs, locally activated lung TRM cells can generate an immediate immune response against invading pathogens [103]. However, recent evidence indicates that exuberant TRM-cell responses contribute to the development of several chronic respiratory conditions, for instance, pulmonary sequelae of post-acute viral infections [104]. Influenza-specific TRM cells produce rapid and robust IFN-γ and TNF-α responses after restimulation in vitro [105]. In human RSV challenge models, the higher frequency of RSV-specific CD8^+^ TRM in BALFs correlate to decreased disease severity and viral load [106]. TRM positioned within the respiratory tract are probably required to limit SARS-CoV-2 spread. During acute SARS-CoV-2 infection, the presence of circulating virus-specific T cell responses, with functional, migratory, and apoptotic patterns modulated by viral proteins and associated with clinical outcome, suggests that a balanced anti-inflammatory antiviral response and long-lasting TRM cells are crucial for protection against SARS-CoV-2 infection [107]. Lung resident γδ T cells represent a major T cell component of mucosal epithelial barrier crucial for maintaining pulmonary homeostasis and influencing the progression of several pulmonary diseases. The γδ T cells are major sources of IL-17A in *K. Pneumoniae* infection and in early host immune defense against acute *Pseudomonas Aeruginosa* pulmonary infection. During *S. pneumoniae* lung infection, a significant increase in the number of activated γδ T cells was observed [108]. The γδ T cell responses to coronavirus infections are still under investigation, with a previous report on SARS-CoV-2 infection showing a strong cytolytic activity against infected target monocytic cell lines [109].

Th-17 cells may be important components of TRM cells. Th-17 cells can elicit serotype-independent immunity to *S. pneumoniae* and *K. pneumoniae* [110,111]. Additionally, the importance of IL-17 has been demonstrated in early host defense against intracellular pathogens such as *Mtb* [112]. Growing evidence suggests that Th17 cells have a crucial role in COVID-19 pathogenesis by boosting cytokine cascade as well as by inducing Th2 responses, inhibiting Th1 differentiation and suppressing Treg cells [113].

Regulatory T (Treg) cells are critical for lung immunological tolerance to airborne allergens and for reducing dangerous immune responses to self- and non-self-antigens [114]. Treg depletion have a role in the pathogenesis of *Chlamydia pneumonia* infection via antigen sensitization [115]. In addition, Tregs are protective against *Pneumococcal* pneumonia, through mechanisms related to TGF-β pathways [116]. Other studies using RSV and influenza A virus mouse models reveal that depletion of Treg cells may result in delayed migration of CD8^+^ T-cell subpopulations [117,118]. Some recent papers describe significantly reduced Tregs numbers in COVID-19 patients and a consequent imbalance in the Treg/Th17 ratio, correlating with a risk of respiratory failure [119].

Lung epithelial cells, AM, and DCs have various pattern recognition receptors (PRRs) in charge of discriminating commensal from pathogenic microbial molecules, both interacting with other receptors and eliciting cytokines and chemokines release [120]. TLRs and NOD like receptors (NLR) belong to the PRR family and are central to balance the activation of downstream signaling and the maintenance of immune tolerance [121]. NLR can regulate inflammatory response and have been involved in lung antibacterial immunity and homeostasis [122]. Furthermore, iterated exposure to pathogen-associated molecular patterns (PAMPs) and damage-associated molecular patterns (DAMPs) from members of the respiratory microbiota induces PRR tolerance in DCs and AM, via the TLRs [123,124]. In conclusion, the respiratory microbiota are interconnected with the airway epithelium and phagocytes in a positive feedback loop to achieve immunological tolerance and avoid uncontrolled inflammatory responses.

The host–microbe interactions in the respiratory tract occur mostly at mucosal sites. Resident microorganisms prime immune cells (epithelial, DCs, and neutrophils) locally or systemically. In fact, these interactions rely on sensing of PRR ligands or metabolites that can enter the circulation and reach other organs. In the gut specifically, this complex link has been extensively studied, but it is still poorly defined in the context of the lung [125,126,127,128]. It is becoming obvious that the respiratory microbiome offers indications to the host immune system that appear to be essential for immune training, organogenesis, and the maintenance of immune tolerance. Numerous observations support the existence of an appropriate time early in life during which correct microbiota perception is essential for immune maturation and consequent respiratory health [8].

For example, members of the *Bacteroidetes* phylum decreased inflammation, neutrophil recruitment, and TLR2-mediated cytokine production compared with *H. influenzae*, in a mouse model [35]. Intranasal inoculation of *Stafiloccoccus aureus* led to monocyte recruitment to the lung that differentiated into AM-dampening IV-induced inflammatory responses [129]. *S. pneumoniae*, although known as a pathogen, is a typical commensal of the upper airways. Simultaneous nasal colonization with *S. pneumoniae* and *H. influenzae* in mice created an inflammatory milieu with high resident abundance of C-X-C motif chemokine ligand 2 (CXCL2) and neutrophils recruitment. The synergistic response depended on production of the pore-forming cytolytic toxin by *S. pneumoniae*, indicating that its presence modulates the immune response to *H. influenzae* [130]. Severe pulmonary disease is present in mice challenged with *H. influenzae.* They display a pronounced access of neutrophils and a high pulmonary abundance of pro-inflammatory cytokines. However, if mice were pre-treated via inhalation with the commensal *Prevotella Nanceiensis*, inflammation was substantially reduced, and tissue pathology absent [35]. Acute intranasal infection of BALB/c mice with adenovirus induced memory AM, characterized by elevated major histocompatibility complex class II (MHCII) expression and transcription of genes related to host defense, chemotaxis, antigen presentation, and glycolytic metabolism. Disease outcome is improved in mice with prior acute adenovirus infection, suggesting that priming of AM by viral infections can generate a persistent trained immunity and improve immune defense against secondary bacterial infection [131].

The role of the microbiome in influencing and managing the host’s immune system and that of the immune system in shaping the microbiome have been studied by means of animal models [9]. Germ-free mice were shown to be more sensitive to infection by *Pseudomonas*, *S. pneumoniae*, and *K. pneumonia* [122,132,133]. Moreover, inoculation of microorganisms in germ-free mice was shown to be essential for DCs recruitment in the lung [134] and the priming of CD8 cells [135]. Antibiotics combinations were also used to assess the role of bacterial microbiota in respiratory infections: generally, these studies have not specifically addressed if oral antibiotics also eliminate lung microbial communities, but such treatments are known to affect upper airways [136]. Mice treated with antibiotics are more vulnerable to viral and bacterial pulmonary infections [137], and antibiotic-treated mice infected with *S. pneumoniae* present with a defect in lung cytokine production [138].

## 3. The Gut–Lung Axis

During dysbiosis caused by a respiratory pathogen, the commensal bacteria is perturbed, and pathobionts can emerge both in the lung and in the gut. Therefore, a disturbance at the level of immune cells potentially leads to tissue damage at both sites. The close physiological and pathological connections between the gut and lung rely mainly on the host–microbe cross-talk [139,140]. Indeed, members of lung and intestine bacteria can directly exchange through blood stream components and metabolites, contributing to health and disease at both sites [137,141] (Figure 2). In the following paragraphs, we will review the most indicative evidence regarding the bidirectional gut–lung axis in the context of bacterial and viral respiratory infections.

### 3.1. Impact of Respiratory Infections on Gut Microbiota

It has been described that alterations of pulmonary microbiota modulate microbial communities of the gut, influencing intestinal signaling [142,143]. Thus, the impact of the lung microbiota on intestinal diseases should be considered. *Mtb* infection is known to cause dysregulation of the immune system, resulting in alteration of the gut microbiome. In one study, comparing the gut microbiome of adult TB patients versus healthy controls, phyla of *Firmicutes*, *Proteobacteria*, and *Verrucomicrobia* were found to be reduced whereas *Actinobacteria*, *Bacteroidetes*, and *Fusobacteria* increased [144]. Another study, analyzing patients with both new and recurrent TB, reported a decrease in *Bacteroidetes*, genus *Prevotella*, and *Lachnospira* and enrichment in *Actinobacteria* and *Proteobacteria* [145]. Finally, in a cohort of affected children, it was observed that there was a depletion of phyla *Actinobacteria* and *Firmicutes*, of genera including *Bacteroides*, *Bifidobacterium*, *Dorea*, *Faecalibacterium*, *Ruminococcus*, and *F. prausnitzii* and an enrichment in *Bacteroidetes*, *Proteobacteria*, *Enterococcus*, and *Prevotella* [146].

The modulation reported in TB patients may lead to a disequilibrium in the production of microbial metabolites, such as short-chain fatty acids (SCFAs), which may reset the lung microbiome and the immune response via the “gut–lung axis”. These findings may also account for the colonization of *Mtb* in the gastrointestinal tract and the development of intestinal TB in pulmonary TB patients [147,148]. In this regard, *F. prausnitzii* is described to have an anti-inflammatory effect, defending against a range of gastrointestinal diseases [149].

Few studies have analyzed the nature of gut microbiota alteration occurring during respiratory viral infections in humans. Although, during influenza, some patients present gastroenteritis-like symptoms despite apparent absence of the virus in the gut.

A study in patients infected with H7N9 virus showed a reduction in phyla *Bacteroidetes* and genus including *Bacteroides*, *Blautia*, *Roseburia*, and *Ruminococcus* but enrichment of *Firmicutes* and *Proteobacteria* and genera, including *Escherichia*, *Clostridium*, and *Enterococcus faecium* [150]. Another study, performed in influenza subtype H1N1 patients, reported a depletion of phyla *Actinobacteria* and *Firmicutes*, and genera including *Dorea*, *Faecalibacterium*, *Ruminococcus*, *Streptococcus* together with an enrichment for both *Actinomycetaceae* and *Micrococcaceae* [151]. In a recent review, 11 different studies reported gut microbiome modifications in patients with a proven or suspected respiratory tract infection (RTI), compared to healthy controls. In summary, gut microbiome alterations in patients were consistently in diversity with a depletion of *Firmicutes*, *Lachnospiraceae*, *Ruminococcaceae* and enrichment of *Enterococcus* [152].

In contrast to the situation with IAV and RSV, viral RNA could be detected in the gut during SARS-CoV-2 infection, even when it was no longer present in the respiratory tract, thus pointing to the digestive tract as potential site of viral replication and activity [153,154]. Gut dysbiosis in SARS-CoV-2 infected patients was associated with COVID-19 disease progress and severity and post-COVID-19 syndrome. It was characterized by decreased anti-inflammatory bacteria like *Bifidobacterium* and *Faecalibacterium* and lowered abundance of butyrate producers such as several genera from the *Ruminococcaceae* and *Lachnospiraceae* families. On the contrary, enrichment of inflammation-associated microbiota, including *Streptococcus* and *Actinomyces*, and overgrowth of opportunistic bacterial pathogens, such as *Streptococcus*, *Rothia*, and *Actinomyces*, were reported [155,156,157]. A recent paper showed that SARS-CoV-2 infection causes gut microbiome dysbiosis in mice, together with alterations of Paneth cells and Goblet cells and markers of barrier permeability. Likewise, microbiome samples collected from 96 COVID-19 patients revealed blooms of opportunistic pathogenic bacterial genera known to include antimicrobial-resistant species, and this gut dysbiosis is associated with secondary bloodstream infections by gut bacteria [158].

Different studies have addressed the effect of respiratory viruses on the gut microbiota using animal models, outlining possible mechanisms by which gut microbiota change during acute viral respiratory infections. In pneumonia induced by *Pneumocystis Murina* (*P. Murina*), the diversity of the intestinal microbial community is severely disturbed: infected mice had altered microbial populations in terms of diversity metrics and relative taxa abundances. Authors also found that mice with CD4^+^ T cell depletion infected with *P. murina* exhibited significantly altered intestinal microbiota that were distinct from infected immunocompetent mice, indicating that the loss of CD4^+^ T cells may also affect the intestinal microbiota in this setting. Interestingly, *P. pneumonia* significantly altered the intestinal microbiota’s potential for carbohydrate, energy, and xenobiotic metabolism as well as signal transduction pathways, thus ultimately affecting the host response to infection [159].

In a mouse model of RSV, an alteration of microbiota diversity was described with increase in *Bacteroidetes* and decrease in *Firmicutes*. This increase in the *Bacteroidetes* phylum was mainly due to a rise in the *Bacteroidaceae*, whereas the reduced abundance of Firmicutes was related to attenuation of both *Lachnospiraceae* and *Lactobacillaceae* families [160]. In influenza A virus (IAV)-infected mice, the signs of intestinal damage and inflammation, altered gene expression, and compromised intestinal barrier functions peaked on day 7 post-infection. As a result of bacterial component translocation, expression of inflammatory markers was upregulated in the liver, and at the same time, an altered gut microbiota composition was observed [161]. A drop was reported in the proportion of *segmented filamentous bacteria* (*SFB*), important in the host resistance against enteric pathogens, paralleled by the emergence of potentially dangerous species, such as *Gammaproteobacteria* and mucus-degrading bacteria. Conversely, infection halted the growth of health-promoting bacteria such as *Lactobacilli*, *Bifidobacteria*, and *SFB* [161].

Respiratory IV can induce intestinal injury that was not caused directly by influenza infection. In fact, no virus was found in the small intestine following intranasal infection, and intra-gastric administration of the virus did not generate intestinal immune alteration. The damage was shown to be mediated by IFN-γ-producing lymphocytes moving, during infection, from the respiratory tract into the intestinal mucosa via the CCL25-CCR9 axis. Consequently, Th17 cells markedly augmented in the small intestine and neutralizing IL-17A was able to decrease intestinal injury. Moreover, antibiotic depletion of intestinal microbiota reduced IL-17A production and attenuated influenza-caused gut damage. Additionally, alteration of intestinal microbiota significantly induced IL-15 production from intestinal epithelial cells, which then promoted Th17 cell polarization directly in the small intestine. In conclusion, these findings provide new mechanistic insights into how respiratory IV infection causes intestinal disease [162].

Adult C57/BL6 mice were exposed to one dose of LPS instillation directly into lungs, and the total bacterial count was significantly increased after 4 and 24 h in the blood and cecum, respectively. Antibiotic treatment reduced the total bacteria in blood but not in the cecum. These data support that lung, blood, and intestinal microbiotas are very dynamic and can be modulated by acute lung injury [163].

Decreased bacteria richness and altered gut microbial composition were observed in mice with hv*Kp* (strain 43816)-induced pneumosepsis, in which *Bifidobacterium* and *Clostridium* were reduced [164]. Wolff et al. employed a mouse model of pneumonia-derived sepsis caused by *K. Pneumoniae* to follow the pathogen burden as well as the composition of the lung, tongue, and fecal microbiota, in the timeframe between local infection and systemic spread. Already at 6 h post-inoculation with *K. pneumoniae*, marked changes in the lung microbiota and differences in the gut microbiota were observed. The gut microbiota was affected by the severity of pneumonia and contributed to the lung microbiota at 12 h post infection [165].

A recent paper shows that cellular immune response to RSV or IV lung infection induces inappetence, which in turn alters the gut microbiome and metabolome. Authors observed that the elimination of CD8^+^ cells prevented the reduction in food intake and inverted the changes in the gut microbiota; this most likely occurred via a secreted TNF-α. Indeed, neutralization of this cytokine during RSV infection reduced the weight loss and attenuated the perturbation of the gut microbiota [166].

Collectively, these studies in animal models pointed to the release of inflammatory cytokines and reduced food intake as possible mechanisms by which acute viral respiratory infections affect the gut microbiota. Interestingly, another mechanism implicated infiltrated CD4^+^ T cells or systemic IFN release, which alters the metabolism of epithelial cells resulting in the accumulation of nutrients, for which the microbes of the intestinal lumen compete. Together with increased oxygen availability, this phenomenon can explain the change from obligate anaerobes to facultative anaerobes such as *Proteobacteria Enterobacteriaceae*. This suggests that hypoxia, a main clinical symptom during the acute phase of respiratory viral infection, could play a role in gut dysbiosis and gastrointestinal disorders during respiratory viral infections [167].

### 3.2. Impact of Gut Microbiota on Respiratory Infection Outcome

The existence of a vital communication between the gut and lung is primarily supported by the evidence of a wide spectrum and severity of pulmonary involvement in patients with IBD [168,169], ranging from subclinical alteration to low-grade and overt chronic inflammatory lung disease [170,171]. Symptoms of airway involvement most often appear in patients with a long-lasting history of IBD [170], thus implicating the dysbiotic gut or the systemically disseminated inappropriate immune response in the pathogenesis. Besides chronic disease, intestinal dysbiosis has been linked to increased susceptibility to respiratory tract infections. In a study involving patients who underwent allogeneic hematopoietic stem cell transplantation (HSCT) and had viral RTIs post transplantation, the number of antibiotic days was associated with progression to LRT disease [172]. Furthermore, clinical observational studies highlighted the importance of healthy gut microbiota in protecting against viral RTIs. Reduced butyrate-producing gut bacteria correlated with increased risk and incidence of viral RTIs in kidney transplant recipients [173], and patients post-allogeneic HSCT [174]. Gut dysbiosis and gut metabolites have been identified in COVID-19 patients correlating with inflammatory response and disease complications [155,175,176,177]. Importantly, fecal transplantation from patients with COVID-19 into germ-free mice caused lung inflammation and worse outcome during pulmonary infection by multidrug-resistant *K. Pneumoniae*, demonstrating that microbiota can directly contribute to disease sequalae [178]. Furthermore, the gut microbiota can also regulate the colonic expression of Angiotensin-converting enzyme 2 (ACE2) receptor [179]. This evidence may contribute to explain the enhanced disease susceptibility and gastrointestinal symptoms in subjects with gut dysbiosis, such as elderly, immune-compromised patients and patients with other co-morbidities [180]. Nonetheless, it must be recognized that the ACE2 receptor is highly expressed by the intestinal epithelia, and it may therefore be involved in the gastrointestinal symptoms [181,182,183].

Absence or depletion of the intestinal commensal bacteria through antibiotics resulted in increased microbial dissemination, inflammation, organ damage, and mortality in several experimental models of bacterial and viral respiratory infections [138,184,185,186]. Most of these effects are related to the ability of gut microbiota to shape systemic immunity. The immune cells and cytokines triggered by gut microbes and their metabolites, such as SCFAs, can reach systemic circulation and regulate the immune and inflammatory responses in the lung, in health and disease [187,188]. This was elegantly demonstrated in the study by Huang et al. in which, after connecting the circulation of two mice, labelled ILC2s from a mouse were found in the lungs of both mice [189]. Remarkably, this did not occur in antibiotic-treated mice [190]. Furthermore, it has been reported that intestinal lymphocytes from IBD patients lack tissue specificity [191]; this may explain the presence of inflammation in extra-intestinal organs in IBD. Dysbiosis-mediated inflammation can also result in increases in circulating levels of fecal calprotectin, plasma C-reactive protein, IL-6, and IL-8, which could contribute to morbidity during lung infection [192,193]. Gut dysbiosis might also impinge on the outcomes of lung infection by reducing nutrient uptake and energy availability, which can in turn disturb the patient’s ability to mount an effective immune response [194]. More mechanistic insights into how gut microbiota influence lung control of respiratory infections come from animal studies. In antibiotic-treated mice, the expression of IFN-γRI, MHC-I, CD86, and CD40 molecules in peritoneal macrophages is blunted during early response to viral infection, suggesting that gut microbiota signal the innate immune response prior to viral replication in the host [195]. Similarly, the establishment of Th1, CTL, IgA, and macrophage response to respiratory viral infections depends on gut microbes. Consistently, rectal TLR stimulation, providing signals for IL-1β and IL-18 secretion, restored lung CD4^+^ and CD8^+^ T cell responses to IV infection in antibiotic-treated mice [186]. An increased mortality due to respiratory viral infection following antibiotic treatment was also related to a decreased abundance of lung Tregs [196]. Overall, these findings confirm that intestinal microbial stimulation is crucial in calibrating the activation threshold for an innate antiviral immune response. Depletion of gut microbiota with antibiotics increased the burden and dissemination of *Mtb* in an experimental model of infection [197]. Particularly, dysbiosis reduced the expression of the innate receptor, macrophage inducible C-type lectin in lung DCs, resulting in impaired stimulatory function towards naïve T cells and reduced effector and memory T cell population in infected mice [198]. Moreover, increased pulmonary colonization by *Mtb* was associated with a significantly reduced accumulation of MAIT cells in the lungs [199]. Remarkably, Ngo et al. recently reported that colonization of the intestine by a single common bacterial species, namely, *SFB*, reprogrammed AM conferring them with enhanced proliferation, complement production, and phagocytosis and resulting in increased protection against IV, RSV, and SARS-CoV-2 [200].

Pulmonary superinfection may represent a further consequence of altered gut microbial composition induced by primary infections in the lung. Increased susceptibility to secondary bacterial infection, particularly those induced by *S. pneumoniae* and *Staphylococcus aureus*, frequently occurs in children and elderly people experiencing respiratory viral infections, leading to morbidity and mortality [201,202]. Sencio et al. demonstrated that gut dysbiosis during influenza contributes to pulmonary pneumococcal superinfection via altered SCFA production. Diminished production of acetate, in mice receiving influenza A virus-conditioned microbiota, altered the bactericidal activity of alveolar macrophages, reduced lung defenses toward secondary pneumococcal infection, and promoted death of superinfected mice [203]. Gut disorders might also contribute to concomitant or secondary bacterial infections in patients with severe COVID-19 [204,205,206]. Besides this evidence, it is conceivable that local lung dysbiosis, altering the dynamics of inter-microbial interactions as well as microbial metabolism, might enhance the proliferation of potentially pathogenic bacterial species.

## 4. Targeting Microbiota to Counteract Respiratory Infections

Unveiling the complex interaction between the lung and gut has been instrumental for a better understanding of the commensal microbiota as a therapeutic target for various kinds of respiratory infectious diseases. Administration of microbes (using probiotics), products favoring their growth (e.g., prebiotics), or microbial metabolites (e.g., postbiotics) can confer host protection during respiratory diseases via direct competition with the pathogenic microbes, improvement of epithelial barrier functions, or immune modulation [207,208,209]. In this section, we report the most relevant examples of microbe-based therapy in the context of human clinical trials and experimental models of respiratory infections.

### 4.1. Microbiota Modulation in the Context of Viral Infections

In a systematic review, Shi et al. reported a series of randomized controlled trials assessing the efficacy of probiotics in preventing viral RTIs in a large cohort of healthy subjects. *Lactobacillus* (L.) was the most used probiotic, followed by *Bifidobacterium* and *Lactococcus*. A majority of these studies showed a reduced risk and incidence of viral RTIs associated with probiotics administration. On the contrary, no consensus was achieved in terms of improvements of clinical manifestations, viral load, and immunological outcomes [210]. The probiotics *L. rhamnosus* and *L. brevis* were also associated with a reduction in the occurrence of influenza infections [211]. The impact of probiotics has been explored in the context of COVID-19. In a retrospective study of ICU patients with SARS-CoV-2-induced pneumonia, treatment with a probiotics cocktail of *Lactobacillus*, *Bifidobacterium*, and *Streptococcus* species showed a positive association with reduced mortality compared to standard care alone [212].

Several reports described the effects of oral administration of probiotics on viral RTIs’ outcomes in animal models, also providing mechanistic insights. Probiotics, such as *Lactobacillus*, *Bifidobacterium*, *Enterococcus*, *or Lactococcus*, administered prior to infection with IV or RSV resulted in mitigation of symptoms and improved survival. Viral load in lungs BALF and nasal washings was also diminished to some extent [210]. Mechanistically, probiotics could elicit protective responses against viral RTIs by engaging immune cells and inducing a specific cytokine/chemokine production profile, though the effects seem to be highly strain-specific [213]. Studies showed increased natural killer (NK) cell activities, decreased infiltrating macrophages and neutrophils, and increased viral specific IgA/G titers in the BALF upon probiotic administration [214]. *L. mucosae* inhibited RSV replication and reduced the proportion of blood inflammatory cells such as granulocytes and monocytes [215]. Jounai et al. reported that, after oral administration in mice infected with murine parainfluenza virus, the probiotic *Lactococcus lactis* was loaded into CD11c^+^ cells in Peyer’s patches and induced type I IFN production by plasmacytoid DCs at mucosal sites. Moreover, increased expression of IFN-related genes in the lungs, suggested that the type I IFN produced by intestinal plasmacytoid DCs could induce a pulmonary anti-viral activity. Consistently, ex vivo stimulation with murine parainfluenza virus of lung lymphocytes from mice treated with the probiotic resulted in high expression of IFN-α and IFN-β [216]. The upregulation of IL-10 and the concomitant reduction in IL-6 during viral infections were also triggered by probiotic administration [217]. In addition to these protective effects, *L. rhamnosus GG* administered intranasally [218] and *L. acidophilus L-92* [219] also increased the levels of IL-1β and monocyte chemotactic protein 1 (MCP-1) cytokines and of the chemokines eotaxin and M-CSF. The mechanisms of probiotics in human RTIs deserve further investigations. Indeed, although probiotics have a satisfactory safety profile, their use could be linked to a higher risk of infection and/or morbidity in frail people [220]. For this reason, there is an increasing interest in the use of non-viable microorganisms [221]. Animal studies proved the beneficial effects of heat-inactivated probiotics in RTIs, although their global effects seemed to be secondary to the live ones.

Gut microbiota can regulate immune responses through production of short-chain fatty acids. Intake of microbially accessible dietary fibers (prebiotics), promoting an increase in the diversity and activity of specific symbiotic microorganisms, leads to variable effects on microbial metabolites production and, in turn, on the host response to infections. A high-fiber (fermentable inulin) diet conveyed protection against influenza through two complementary mechanisms. Fed mice exhibited enhanced bone marrow generation of alternatively activated macrophages with a limited capacity to promote CXCL1-mediated recruitment of neutrophil to the airways, thus leading to limited tissue immunopathology during infection. In parallel, diet-derived SCFAs stimulated CD8^+^ T cell antiviral activity. Such an effect was mediated by butyrate through the free fatty acid receptor (FFAR)3 [208]. Likewise, a high-fiber (fermentable pectin) diet protected against RSV infection by stimulating type I IFN response in lung epithelial cells. The protection was mediated by acetate via GPR43 [222]. Furthermore, Sencio V et al. reported that oral acetate supplementation during influenza infection reinforced, in a FFAR2-dependent manner, the lung defenses against secondary pneumococcal infection and reduced the lethal outcome of superinfected mice [203]. Similarly, intranasal acetate increased interferon-dependent responses and reduced lung viral load during rhinovirus infection [223]. Desaminotyrosine, produced by an obligate clostridial anaerobe (*Clostridium orbiscindens*) from digestion of a flavonoid-enriched diet, can reach the lungs and prime the immune system to protect from influenza infection. In this setting, desaminotyrosine amplified type I IFN signaling in phagocytes via IFN-α/β receptor and STAT1 [185]. Interestingly, a trial evaluating the efficacy of prebiotics (galacto-oligosaccharide and polydextrose) in preventing viral RTIs in newborns demonstrated that prebiotics displayed a superior beneficial effect compared to probiotics, likely related to the direct stimulatory effect on the growth of pre-existing good bacteria [224].

### 4.2. Microbiota Modulation in the Context of Bacterial Infections

The use of probiotics has also been attempted in different clinical trials [225]. A majority of these studies analyzed the efficacy of probiotics for prevention and treatment of nosocomial pulmonary infections in ICU patients. In this setting, the probiotics *L. casei rhamnosus* and *L. rhamnosus*, administered orally or oropharyngeally, resulted in decreased colonization and infection of the LRT by *Pseudomonas aeruginosa* or related pathogens [226,227]. One study observed a decreased incidence of ventilator-associated pneumonia (VAP) in ICU patients with sepsis, upon administration of a cocktail of *B. breve* Yakult, *L. casei Shirota*, and galactooligosaccharides [228]. Studies in mice infected with *S. pneumoniae* suggested that oral administration of different probiotics, such as strains from *Lactobacillus* and *Streptococcus* genera, caused increased resistance to infection, decreased pulmonary bacteria load, and increased survival [229,230,231]. Protection was associated with increased lung infiltrations of neutrophils, macrophages, and lymphocytes, as well as a higher titer of anti-*S. pneumoniae* IgG and IgA. Comparable results were observed in mice infected with *P. aeruginosa* [232,233]. In addition, the use of *L. rhamnosus* potentiated the anti-inflammatory response by increasing Foxp3^+^ Tregs and decreasing proinflammatory IL-6. Such an anti-inflammatory profile was also observed in infected mice following intratracheal administration of other *L.* strains [234]. The administration of viable or inactivated probiotic *Bifidobacterium longum 51A* stimulated lung clearance of *K. pneumoniae* by enhancing ROS production in alveolar macrophages and reducing pro-inflammatory TNF-α and IL-6. However, only viable probiotics induced a concomitant increase in IL-10 levels, primarily mediated by acetate [235]. Similarly, acetate (SCFAs) supplementation reduced inflammatory infiltrates in lung parenchyma and TNF-α and IL-1β levels induced by *K. pneumoniae* infection as well as restored bactericidal activity of alveolar macrophages in respiratory pneumococcal infection, secondary to IFV [236]. Intranasal or oral inoculation of bacterial strains potently activating Nod2 receptors (*L. crispatus*, *Staphylococcus aureus* and *Staphylococcus epidermidis*, or *L. reuteri*, *Enterococcus fecalis*, *L. crispatus*, and *Clostridium orbiscindens*, respectively) protected mice against *S. pneumoniae* or *K. pneumoniae* infections, based on the ability of these strains to stimulate the production of GM-CSF [122]. Supplementation with *Lactobacillus* could also restore dendritic-cell-mediated anti-*Mtb* immunity in the lungs [198]. Oral treatment with *A. muciniphila* or *A. muciniphila*-mediated palmitoleic acid strongly inhibited tuberculosis infection through epigenetic inhibition of TNF in mice infected with *Mtb* [237].

While most commensal bacteria used as probiotics originate from the intestine, attempts have been made to use respiratory commensal bacteria. Particularly, intranasal administration in infant mice of *Corynebacterium pseudodiphteriticum* was able to ameliorate features of both RSV primary infection and of secondary *S. pneumoniae* superinfection, lowering both pathogen burden and lung damage [238]. To our knowledge, no human studies to date have assessed the potential for respiratory probiotics (i.e., viable microbiota instilled or aerosolized into the lower respiratory tract). To achieve this, practical issues need to be addressed and overcome [239].

Another potential approach to neutralize respiratory pathogens is the use of so-called “predatory bacteria” from the genera *Bdellovibrio* spp. or *Micavibrio* spp., directly killing other Gram-negative bacteria [240]. Intranasal administration of both *B. bacteriovorus* and *M. aeruginosavorus* considerably reduced, in animal models, respiratory *K. pneumoniae* burden. Their subsequent elimination by the host-innate immune mechanisms, with no adverse effect, envisions their possible application for the treatment of bacterial pneumonia in humans [241,242].

Selective depletion of opportunistic bacterial pathogens achieved with monoclonal human antibodies (mAbs), targeting and inactivating bacteria and their virulence factors and/or toxins, is one of the most promising antibiotic-independent approaches to fight infectious illnesses [243]. Indeed, owing to their target specificity, they do not exhibit adverse effects on the indigenous microbiota and are unlikely to induce widespread resistance. Furthermore, polyvalent mAbs are engineered to exert multiple anti-bacterial actions, including virulence factors inactivation as well as complement deposition and activation of innate immunity [244,245]. Gremubamab (MEDI3902; AstraZeneca) is a bispecific human IgG1 mAb that selectively binds to the *P. aeruginosa* virulence factors and was developed to prevent nosocomial pneumonia in high-risk patients [246]. The mAb promotes bacterial clearance by neutrophils and prevents its attachment to airway epithelial cells [247]. Prophylactic as well as therapeutic administration of Gremubamab was highly protective in animal models of acute *P. aeruginosa* pneumonia and of VAP [248]. A clinical trial in *P. aeruginosa*-colonized ICU patients showed a risk reduction in patients with lower baseline inflammation. The monoclonal human IgM antibody Panobacumab (AR-101, Aerumab; Aridis Pharmaceuticals) is in clinical development for counteracting *P. aeruginosa* in hospital-acquired pneumonia [1,249]. Panobacumab targets LPS from the highly prevalent *P. aeruginosa* serotype O11, contributing to most nosocomial pneumonia cases. Administration of Panobacumab reduced bacterial burden and mitigated lung inflammation in a mouse model of acute *P. aeruginosa* infection [250]. Two other antibodies have been tested for the prevention and treatment of *S. aureus* pneumonia. Suvratoxumab (MEDI4893; AstraZeneca, outlicensed to Aridis Pharmaceuticals) specifically binds to and inactivates the pore-forming alpha-toxin of *S. aureus*, a highly conserved key virulence factor [251]. In preclinical studies, Suvratoxumab was shown to confer protection [252]. In clinical trials on mechanically ventilated ICU patients colonized with *S. aureus*, the mAb showed a significant reduction in pneumonia and in the duration of hospitalization and ICU stay in a subset of patients [253]. Several other antibacterial human mAbs are currently in preclinical and early clinical development, including AR-401 (Aridis Pharmaceuticals) and ASN-5 (Arsanis, outlicensed to BB200) targeting *A. baumannii* and *K. pneumoniae*, respectively [245].

Phage therapy has received clinical interest to treat respiratory infections with MDR bacterial pathogens, because of its ability to selectively eliminate target bacteria without impacting the host microbiota, compatibility with antibiotics, and low immunogenicity [254]. Phage formulations show strong preclinical efficacy in the treatment of *P. aeruginosa*, *K. pneumoniae*, *A. baumannii*, or *E. coli* and are nowadays being evaluated in the clinic for the treatment of human respiratory infections [255,256]. A mix of four obligate lytic bacteriophages targeting *P. aeruginosa* respiratory infections was used to successfully treat a patient with VAP pneumonia and emphysema [257]. A second-generation phage cocktail, AP-PA02 (Armata Pharmaceuticals), has been recently evaluated in individuals with chronic *P. aeruginosa* lung infections, and final trial results are awaited (https://clinicaltrials.gov/study/NCT04596319 (accessed on 15 January 2024). Phage-derived endolysins or engineered lysins may display advantages over the use of whole bacteriophage preparations to combat bacterial infections, as resistance is hard to develop due to the conserved nature of their cell wall targets. Administration of phage endolysins diminished *S. pneumoniae* titers in a mouse model of nasopharyngeal colonization and protected mice from fatal pneumococcal pneumonia or *P. aeruginosa* infection, respectively, suggesting their potential to prevent and treat respiratory bacterial infections [258,259].

## 5. Concluding Remarks

RTI is one of the most common infectious diseases of viral or bacterial origin, causing both social and economic burden. The importance and complex cross-talk between the lung and the gut and the strong link of the gut–lung axis with respiratory health are increasingly recognized. The cross-talk along this gut–lung axis is mediated by transfer of microbes or microbial and host metabolites in the blood from one tissue to the other and by the immune system (Figure 3).

However, the understanding of the mechanisms involving the gut–lung axis, particularly in the context of respiratory infections has just begun. The interactions between the microbiome, the respiratory mucosa, and the underlying pathways need to be better understood. One question to be addressed is whether respiratory and gut microbiota influence distinct aspects of lung immunity. Furthermore, better defining how microbiota’s composition and function is affected during infections will further clarify their pathogenesis and development. Unfortunately, the substantial understanding of microbe modifications in patients with a specific RTI remains incomplete, due to technical shortcomings and the small number of studies available for each respiratory pathogen. On the other hand, they may also suffer from important limitations, linked to the use of medication (antibiotics and anti-virals) to treat the disease. Similarly, it may be difficult to differentiate respiratory manifestations of intestinal disorders from respiratory symptoms or complications related to their treatment [260].

For this reason, a majority of human studies are still observational and do not directly prove whether disease progression is led by a changed microbiome. In this regard, robust animal models will be crucial to address key questions, such as the causal relationship and, eventually, the specific mechanisms by which the microbiome affects the host.

If extensive research is available on the effect of gut microbial metabolites on disease and immunity, the impact of lung microbiome metabolites on physiology and disease is still largely unexplored. Understanding if they differ from those produced in the gut and their effects on respiratory mucosa and immune cells would be instrumental for the identification of possible biomarkers of disease progression, disease stratification, or treatment.

Finally, even if the great majority of the lung and gut microbiota are composed of bacteria, fungi and viruses are also part of it [143] and might have an impact on host immunity and pulmonary inflammation during infections [261].

Since the gut microbiota plays a key role in influencing the immune system, it affects both local and systemic (in the lung) responses to pathogens. Accordingly, microbiota-targeted therapies that modify the gut microbiome have been shown to benefit both acute and chronic respiratory conditions in animal models. Nevertheless, evidence that intake of probiotics or postbiotics results in improvement of respiratory conditions in clinical setting is still limited [262]. The most used probiotics in studies of pulmonary infections, both caused by viruses and bacteria, are those from the *Lactobacillus* genus. Despite this evidence, the mechanistic basis for the observed beneficial effects is often not well defined, due to differences in study design and experimental protocols. Therefore, more in-depth investigations should be carried out, with well-defined experimental protocols, to better understand the function of probiotics in the immunity against pulmonary infections.

To date, most commensal bacteria used as probiotics are of gastrointestinal origin. In many aspects, exploitation of gut microbiome for therapeutic purposes is compelling: it is more accessibly sampled and studied, and it is greater in biomass and metabolic activity and easier to be modulated. However, recent studies have revealed that lung microbiota are consistently more closely correlated with variation in lung immunity than are gut communities [263,264]. Moreover, experimental modulation of lung commensals directly and persistently alters the lung immune response [14,64], suggesting that they may have potential as locally applied probiotics for the prevention and management of respiratory infections.

Finally, it is worth mentioning the problem of disease-causing bacteria resistant to antibiotics [265]. LRT infections are the leading cause of mortality among all multidrug-resistant infections and are often associated with the priority pathogens *S. aureus*, *K. pneumoniae*, *S. pneumoniae*, *Acinetobacter baumannii*, and *P. aeruginosa*. Likewise, respiratory viral infections are becoming increasingly difficult to manage due to the emergence of novel variants. For this reason, development of novel preventive and therapeutic strategies is desirable. Owing to the critical role of lung and gut dysbiosis in lower airway disease, manipulation of the microbiome might be one possible intervention avenue.

## Figures and Tables

**Figure 1 ijms-25-04051-f001:**
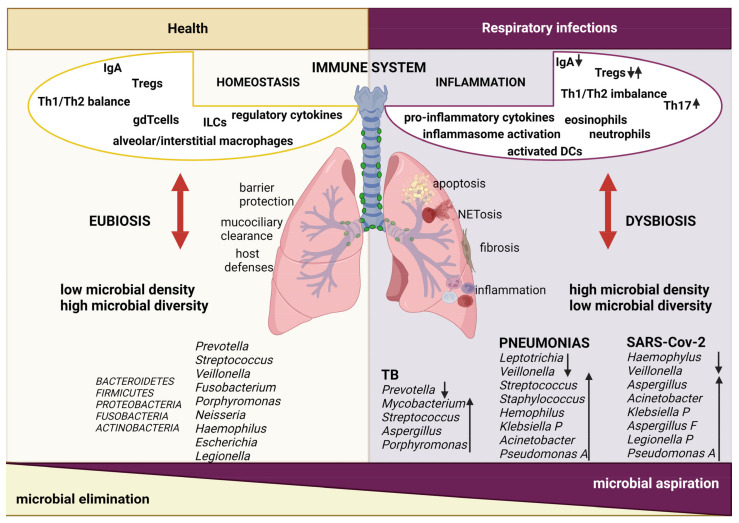
Schematic overview of the lung microbiota and immunity in health and during respiratory infections. The lung microbiota plays a critical role in lung homeostasis. In eubiotic condition, Proteobacteria, Firmicutes, Fusobacteria, Bacteroidetes, and Actinobacteria phyla mainly colonize the human lungs. Lung microbiota can promote the polarization of naïve T cells and the differentiation of alveolar macrophages to protect against pathogens. Moreover, mucociliary clearance and barrier protection are fundamental mechanisms together with host defenses. On the other hand, dysbiosis of the lung microbiome leads to immune cells activation. Immune cells then migrate into the tissue, produce proinflammatory cytokines, and finally contribute to local inflammatory response. Moreover, alterations in cytokines milieu promote pathologic fibrotic remodeling, NETosis, and apoptosis. Arrows indicate changes in relative abundance of immune cell subsets and bacterial species. Credit: BioRender.com.

**Figure 2 ijms-25-04051-f002:**
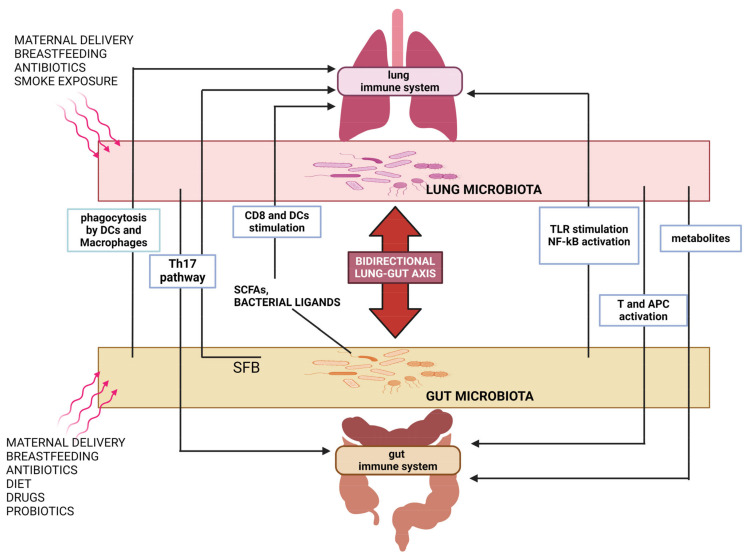
Schematic representation of the main interactions underlying the gut–lung axis. The interactions between the gut and the lung are mediated by the microbiota and its products as well as by the immune cells. A bidirectional communication is recognized: *SFB*, a commensal gut microbiota, and microbial metabolites, such as SCFAs, stimulate and promote the differentiation of Th-17 cells, which have immunomodulatory functions in the lungs. Moreover, the gut microbes enter the intestinal mucosa and may be phagocytosed by antigen presenting cells (APCs), DCs, and macrophages. Travelling to the lung, APCs stimulate T cells and lung immune responses. On the other hand, the lung microbiota exhibits similar effects influencing the immune system and homeostasis of the gut. Several factors are well known to influence the composition of the intestinal and/or lung microbiota, such as diet, drugs, etc. Credit: BioRender.com.

**Figure 3 ijms-25-04051-f003:**
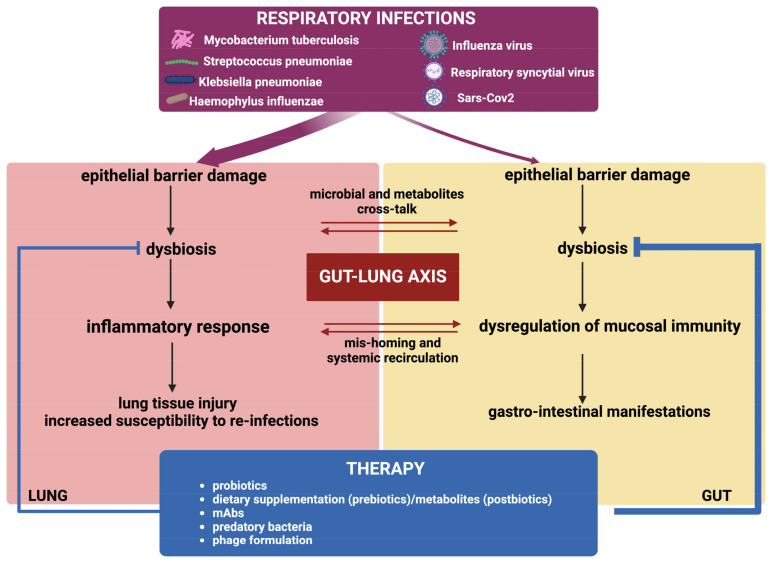
Summary of the main topics and mechanisms addressed in the present review. Respiratory infections directly affect lung epithelial barrier integrity (bold arrow), while their effects on gut permeability are indirect, except for SARS-CoV-2 which has been demonstrated to infect intestinal epithelial cells (thin arrow). This results in a cascade of events ending with pathological manifestations. The different therapy attempts are designed to inhibit gut dysbiosis (bold inhibitory arrow) with the consequent immune modulation and host protection. Only a minority of these therapies target directly the lung microbiota (thin inhibitory arrow). Credit: BioRender.com.

## Data Availability

No new data were created or analyzed in this study. Data sharing is not applicable to this article.

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
