# Peer review of "Microbiota and Immunity during Respiratory Infections: Lung and Gut Affair"

_ijms, 2024, doi:10.3390/ijms25074051_

Round 1
Reviewer 1 Report
Comments and Suggestions for Authors
The review report is attached

Author Response
We thank the Reviewers for their helpful comments on our work. We have revised our manuscript and incorporated new discussion sections in order to address all the Reviewersʼ suggestions. To aid the Reviewers in the re-evaluation of this study we included a marked copy of our revision showing the changes made.
Reviewer #1
The manuscript reviewed the “Microbiota and Immunity during respiratory infections.” Gut-lung axis. This is a very comprehensive review.
Comments
1.Abstract a. Include a line on future emphasis.
As suggested by the reviewer we have included in the abstract a final sentence that refers the reader to an outlook by the authors on the most important research points to be carried out in the future in the field.
- a. We corrected the typo at original line 32.
- b. What is the mechanism by which lung infections affect the gut microbiota?
To aid the reader in the understanding of the mechanisms by which lung infections affect the gut microbiota, we have recapitulated the main known pathways involved at the end of the dedicated paragraph 3.1.
- c. Which strains (gram-negative or gram-positive) bacteria are responsible for the infection?
Following the reviewer’s suggestion, we have specified in the paragraph 2.2 “Lung microbiota modifications during respiratory infections” if bacteria responsible for the infectious diseases are gram positive or negative.
- d. Emphasize the colony size required for immunity.
We apologize with the reviewer, but for us it is a bit unclear what this means (sociality? diversity of microbiota? pathogen burden?).
- e. Emphasize animal studies.
As indicated by the reviewer, we have better underlined in the text (lines 561-570; 861-865) the importance of animal studies in the mechanistic insights related to the gut-lung axis.
- f. How do gender, BMI, and diet affect the immunity?
We have mentioned these factors influencing lung microbiota and immunity at lines 146-148 (paragraph 2.1.) and indicated relative references.
- g. How does the treatment of lung infections affect the microbiota?
We agree with the reviewer that use of medication (antibiotics and anti-virals) for the treatment of lung infection, likely causing alteration in the microbiota composition, might affect the interpretation of the results in the human studies. To overcome this limitation, robust animal models will be crucial in future research. We have highlighted this issue in the conclusive remarks (lines 856-860).
- h. How did the COVID vaccine affect the gut and lung microbiota?
We thank the reviewer for raising this interesting point. Indeed, effects of COVID vaccines on gut microbiota is currently a hot topic of research. However, we think that this specific aspect is out of the scope of our review, which is instead focused on the therapies targeting specifically the host microbiota.
Reviewer 2 Report
Comments and Suggestions for Authors
The review article titled "Microbiota and immunity during respiratory infections: lung and gut affair" by Marrella and Cassani gave an overview of the lung microbiome modifications occurring during respiratory infections and of the functional mechanisms underlying host−microbe interactions.
The review article is sound, and some minor comments need to be considered as follow:
- For figure 1, please transfer it after its mentioning in the text.
- Please revise the sentence in lines 206-209.
- Some scientific names need to be revised in the manuscript as the name of some species starts with capital letters.
- All abbreviations are mentioned in the manuscript text. I think it is not needed to be repeated before references.
- Please carefully revise the manuscript for some minor grammar errors.
Comments on the Quality of English Language
Minor revision of English language is required.
Author Response
We thank the Reviewers for their helpful comments on our work. We have revised our manuscript and incorporated new discussion sections in order to address all the Reviewersʼ suggestions. To aid the Reviewers in the re-evaluation of this study we included a marked copy of our revision showing the changes made.
Reviewer # 2
The review article titled "Microbiota and immunity during respiratory infections: lung and gut affair" by Marrella and Cassani gave an overview of the lung microbiome modifications occurring during respiratory infections and of the functional mechanisms underlying host−microbe interactions. The review article is sound, and some minor comments need to be considered.
- For figure 1, please transfer it after its mentioning in the text.
We thank the reviewer for the suggestion, we have mentioned the Figure 1 also in the paragraph 2.1
- Please revise the sentence in lines 206-209.
We corrected it as follows: “the disease severity and mortality rate differed based on age comorbidities, many of which were linked to gut and lung microbial alterations. This suggests that dysbiosis could predict COVID-19 severity”
- All abbreviations are mentioned in the manuscript text. I think it is not needed to be repeated before references.
We thank the reviewer for the suggestion, we can exclude the final abbreviations if all reviewers and the editor agree.
Reviewer 3 Report
Comments and Suggestions for Authors
This abstract presents an intriguing overview of the current understanding and research gaps regarding the interaction between the gut microbiota and lung health, particularly in the context of respiratory tract infections. However, for submission to IJMS, several improvements could enhance clarity, coherence, and impact. Here are some suggestions:
-
Clarify and Streamline the Introduction: The opening sentence could be more precise about the specific types of respiratory tract infections being discussed. Clarifying whether the focus is on viral, bacterial, or all types of infections could help narrow the scope and relevance of the review.
-
Specificity in Language: The phrase "gut microbial dysbiosis is believed to be associated with disease etiology or/and development in the lung" is somewhat vague. It would be beneficial to specify whether there is direct evidence for this association, the types of diseases most affected, and whether the relationship is correlative or causal.
-
Detail on Mechanisms: The statement "mechanisms by which the lung impacts the intestinal environment have not yet fully identified" is intriguing but lacks detail. Providing examples of known or hypothesized mechanisms—even if not fully understood—would enrich the review's context.
-
Enhance the Focus on the Lung Microbiome: While the abstract mentions the lung microbiome modifications during respiratory infections, it stops short of detailing these modifications. A brief overview of key changes and their potential implications for health and disease would be valuable.
-
Clarify the Scope of the Review: The abstract suggests a broad coverage, from microbial modifications to therapeutic possibilities. Ensuring that the review comprehensively covers each of these areas—or specifying if the focus is more on one aspect than others—would help set reader expectations.
-
Evidence for Gut-Lung Axis: The discussion on gut-lung bidirectional communications is central to the review. Strengthening this section with current evidence, including animal models or human studies that support this communication, would significantly enhance the review's authority.
-
Future Directions and Therapeutic Possibilities: While the abstract mentions challenges and possibilities related to novel therapies, providing a more detailed discussion or examples of what these might entail (e.g., probiotics, fecal transplants, dietary interventions) could make the conclusion more impactful.
-
Technical and Stylistic Adjustments:
- Consider removing hyphens that are not grammatically necessary (e.g., "host−microbe interactions" should be "host-microbe interactions").
- Ensure consistent use of the Oxford comma for clarity.
- Revise any instances of passive voice that could be more directly stated in the active voice to enhance readability.
-
Conclusion: The abstract should conclude with a strong statement on the importance of the review's findings or insights, emphasizing the potential impact on the field and future research directions.
For approval, I suggest an extensive and major review of the study.
Summarize the text into 5 main topics:
Throughout the text, there are extremely long subtitles, as if the publications were extensively placed to extinguish the subject, making reading overwhelming and time-consuming. I suggest the following changes:
a) Divide the study into 5 main topics, with a maximum of 10 paragraphs per topic in each title.
b) make a constellation graph, with the most frequent words in the study, illustrating the key theme.
c) Make a new illustration, using biorender, illustrating the broad mechanism to which the publication refers.
d) Review English with a native speaker, there are several errors making the text sometimes incomprehensible
Comments on the Quality of English Language
Low quality of english. Native speaker could help.
Author Response
We thank the Reviewers for their helpful comments on our work. We have revised our manuscript and incorporated new discussion sections in order to address all the Reviewersʼ suggestions. To aid the Reviewers in the re-evaluation of this study we included a marked copy of our revision showing the changes made.
Reviewer #3
This abstract presents an intriguing overview of the current understanding and research gaps regarding the interaction between the gut microbiota and lung health, particularly in the context of respiratory tract infections. However, for submission to IJMS, several improvements could enhance clarity, coherence, and impact.
- Clarify and Streamline the Introduction: The opening sentence could be more precise about the specific types of respiratory tract infections being discussed. Clarifying whether the focus is on viral, bacterial, or all types of infections could help narrow the scope and relevance of the review.
As suggested by the reviewer we better specified both in the Abstract and in the Introduction the bacterial and viral infections discussed in this work.
- Specificity in Language: The phrase "gut microbial dysbiosis is believed to be associated with disease etiology or/and development in the lung" is somewhat vague. It would be beneficial to specify whether there is direct evidence for this association, the types of diseases most affected, and whether the relationship is correlative or causal.
We better clarified the abstract accordingly. In addition, the issue of correlative versus causal relationship between gut microbiota and lung disease has been addressed along the description of the experimental evidences, and in the conclusive remarks.
- Detail on Mechanisms: The statement "mechanisms by which the lung impacts the intestinal environment have not yet fully identified" is intriguing but lacks detail. Providing examples of known or hypothesized mechanisms—even if not fully understood—would enrich the review's context
As requested, we clarified the statement in the Abstract, introduced a summary paragraph in the Introduction and extensively described known and hypothesized mechanisms in the paragraph 3.1.
- Enhance the Focus on the Lung Microbiome: While the abstract mentions the lung microbiome modifications during respiratory infections, it stops short of detailing these modifications. A brief overview of key changes and their potential implications for health and disease would be valuable.
We modified the abstract according with words’ number constrains.
- Clarify the Scope of the Review: The abstract suggests a broad coverage, from microbial modifications to therapeutic possibilities. Ensuring that the review comprehensively covers each of these areas—or specifying if the focus is more on one aspect than others—would help set reader expectations.
We think that the topics highlighted in the abstract are consistently addressed in the text.
- Evidence for Gut-Lung Axis: The discussion on gut-lung bidirectional communications is central to the review. Strengthening this section with current evidence, including animal models or human studies that support this communication, would significantly enhance the review's authority.
Chapter 3 is entirely dedicated to discuss the experimental evidence supporting this communication.
- Future Directions and Therapeutic Possibilities: While the abstract mentions challenges and possibilities related to novel therapies, providing a more detailed discussion or examples of what these might entail (e.g., probiotics, fecal transplants, dietary interventions) could make the conclusion more impactful.
We modified the abstract as suggested.
- Technical and Stylistic Adjustments:
- Consider removing hyphens that are not grammatically necessary (e.g., "host−microbe interactions" should be "host-microbe interactions").
- Ensure consistent use of the Oxford comma for clarity.
- Revise any instances of passive voice that could be more directly stated in the active voice to enhance readability.
As suggested, we revised the text accordingly with the advice of a native speaker.
- Conclusion: The abstract should conclude with a strong statement on the importance of the review's findings or insights, emphasizing the potential impact on the field and future research directions.
As suggested by the reviewer we have included in the abstract a final sentence that refers the reader to an outlook by the authors on the most important research points to be carried out in the future in the field (conclusive remarks).
I suggest the following changes:
a) Divide the study into 5 main topics, with a maximum of 10 paragraphs per topic in each title.
We have divided the review in 5 main topics as suggested by the reviewer.
b) make a constellation graph, with the most frequent words in the study, illustrating the key theme.
We are not sure to have understood what the reviewer means with constellation graph. However, to meet the reviewer request, we added a New Figure 3, schematically resuming all the topics, and underlined mechanisms, addressed in the review.
Reviewer 4 Report
Comments and Suggestions for Authors
The authors set as their objective a study of the specialized literature on the bidirectional gut-lung communication in mediating immunity and inflammation. At the same time, they propose to evaluate the opportunities regarding the therapeutic approach based on microbiota.
After careful analysis of the review, we found the following:
The authors provide only an overview of the physiological mechanisms of lung defense in bacterial and viral respiratory infections. These aspects are presented descriptively in sections such as:
- Lung microbiota – the conclusion is that the lung microbiome and the interaction between commensal microbial communities and lung immunity is just in its infancy and future studies are needed to deeply investigate the specific mechanisms.
- Changes in the lung microbiota during respiratory infections – which highlights the diversity of the respiratory microbiota and the fact that it remains an insufficiently debated issue.
- Lung defense during respiratory infections – in which the authors conclude that respiratory microbiota interacts with airway epithelium and phagocytic cells in a positive feedback loop to develop immunological tolerance and prevent exaggerated inflammatory responses.
- Gut-lung axis - the authors highlight the fact that a disturbance in immune cells can lead to tissue damage in both places. It is mentioned that this aspect is due to the close physiological and pathological connections between the intestine and the lungs.
- In section 6, a series of aspects regarding the targeting of the microbiota to counteract viral and bacterial respiratory infections are presented descriptively.
Finally, the authors conclude that the understanding of the mechanism involving the gut-lung axis is still in its infancy and remains to be further elucidated.
Remarks:
Although the presentation of the results of a review study does not require compliance with a protocol such as PRISMA, it is still preferable to have some clarity in the presentation of the study's methodology.
Thus, it would have been necessary for this review to present the eligibility criteria of the studies included for analysis in order for the conclusions obtained to be reproducible. It is also important to include the most current research, for example research published in the last five years.
The authors do not make any reference to these aspects. For this reason the article is a narrative one without significant relevance.
For example in section 7. Concluding remarks:
The authors present a series of arguments based, as they mention, on recent studies, but these are from the years 2015-2017.
Rows: 791 – 794:
“Finally, even if the great majority of the host microbiota is composed of bacteria, recent evidence has demonstrated that fungal agents can also have beneficial functions and are a natural part of the lung microbiota [254-256].”
A conclusion regarding the benefits of probiotic and prebiotic therapies bringing into discussion "preclinical models" (rows: 797 - 799).
At the same time in the following lines (rows: 799 - 800) the authors mention that there is scientific evidence that supports that the benefits of these therapies are still limited (“Accordingly, probiotic and prebiotic therapies that modify the gut microbiome have been shown to benefit both acute and chronic respiratory conditions, in preclinical models.”).
Considering the type of article, I believe that these conclusions should be rigorously referenced. In many paragraphs the bibliographic references are not mentioned.
Authors do not adhere to these minimum rigors. We also noted that of the 259 bibliographic references, approximately 75% are older than five years.
In this context, I recommend redoing the manuscript and respecting some rigors of presenting the research results so that they clearly answer the research question.
Comments on the Quality of English LanguageMinor editing of English language required
Author Response
We thank the Reviewers for their helpful comments on our work. We have revised our manuscript and incorporated new discussion sections in order to address all the Reviewersʼ suggestions. To aid the Reviewers in the re-evaluation of this study we included a marked copy of our revision showing the changes made.
Reviewer #4
The authors set as their objective a study of the specialized literature on the bidirectional gut-lung communication in mediating immunity and inflammation. At the same time, they propose to evaluate the opportunities regarding the therapeutic approach based on microbiota…..
Remarks:
Although the presentation of the results of a review study does not require compliance with a protocol such as PRISMA, it is still preferable to have some clarity in the presentation of the study's methodology.
Thus, it would have been necessary for this review to present the eligibility criteria of the studies included for analysis in order for the conclusions obtained to be reproducible. It is also important to include the most current research, for example research published in the last five years.
The authors do not make any reference to these aspects. For this reason the article is a narrative one without significant relevance.
For example in section 7. Concluding remarks:
The authors present a series of arguments based, as they mention, on recent studies, but these are from the years 2015-2017.
Rows: 791 – 794:
“Finally, even if the great majority of the host microbiota is composed of bacteria, recent evidence has demonstrated that fungal agents can also have beneficial functions and are a natural part of the lung microbiota [254-256].”
A conclusion regarding the benefits of probiotic and prebiotic therapies bringing into discussion "preclinical models" (rows: 797 - 799).
At the same time in the following lines (rows: 799 - 800) the authors mention that there is scientific evidence that supports that the benefits of these therapies are still limited (“Accordingly, probiotic and prebiotic therapies that modify the gut microbiome have been shown to benefit both acute and chronic respiratory conditions, in preclinical models.”).
Considering the type of article, I believe that these conclusions should be rigorously referenced. In many paragraphs the bibliographic references are not mentioned. Authors do not adhere to these minimum rigors. We also noted that of the 259 bibliographic references, approximately 75% are older than five years.
In this context, I recommend redoing the manuscript and respecting some rigors of presenting the research results so that they clearly answer the research question.
As requested by the reviewer, we revised our manuscript including the most current research references. However, older references were also cited if the authors think they have represented a fundamental breakthrough in the field. In addition, thanks to the constructive suggestions provided by the other 4 reviewers, we implemented the overall quality of our manuscript by strengthening our conclusions and providing technical and stylistic adjustments. Particularly, the abstract, the introduction and the conclusive remarks were extensively revised. A new Figure 3, schematically resuming all the topics, and underlined mechanisms, addressed in the review was included.
Finally, we would like to point out that the intent of the authors with this manuscript was to produce a narrative review, providing a broad perspective on a subject, namely the link between microbiota and immunity during respiratory infections with focus on the role of gut-lung axis in this context. This type of review differs from a systematic review, which starts with clear question to be answered or hypothesis to be tested. In addition, explicit references search strategy or methodology are not usually specified. On the other hand, searching for relevant studies may be potentially biased by the general opinion/perspective of the author.
Reviewer 5 Report
Comments and Suggestions for Authors
In the timely review, the authors have discussed the importance of gut microbiome in regulating host immunity and its association with disease etiology and progression in lungs leading to respiratory infections. The review comprehensively describes various targeted treatment approaches and their associated challenges in development towards respiratory disease’s treatment.
The review manuscript is well written, and after preliminary proof read from the authors for minor punctuations, should be accepted in the current form.
Author Response
Reviewer #5
In the timely review, the authors have discussed the importance of gut microbiome in regulating host immunity and its association with disease etiology and progression in lungs leading to respiratory infections. The review comprehensively describes various targeted treatment approaches and their associated challenges in development towards respiratory disease’s treatment.
The review manuscript is well written, and after preliminary proofread from the authors for minor punctuations, should be accepted in the current form.
We thank the reviewer for her/his appreciation of our work.
Round 2
Reviewer 4 Report
Comments and Suggestions for Authors
The authors made important additions to the manuscript thus improving its quality.
I recommend removing the phrase written in line 868-869.
I believe that the authors have mentioned the most important views in their study, and possible omissions do not require this mention: „We apologize to those investigators whose work we were unable to cite for space limitations.”
Comments on the Quality of English LanguageFine proofreading of the English language is required.
Author Response
We thank the Reviewer for her/his comment to our revised manuscript. As suggested, we removed the final sentence and edited the manuscript for english language.